# Reproductive health programs in women with physical disabilities: A scoping review protocol

**Maryam Abedi**[1], **Mehran Soleymani**[2], **Seyedeh Solmaz Talebi**[3], **Zahra Motaghi**[4]*

1 Student Research Committee, School of Nursing and Midwifery, Shahroud University of Medical Sciences, Shahroud, Iran, 2 Department of Psychology, Azarbaijan Shahid Madani University, Tabriz, Iran, 3 Department of Epidemiology, School of Public Health, Shahroud University of Medical Sciences, Shahroud, Iran, 4 Department of Reproductive Health, School of Nursing and Midwifery, Shahroud University of Medical Sciences, Shahroud, Iran

* zhrmotaghi1@gmail.com

## Abstract

### Introduction

Approximately 15% of people with disabilities are women of reproductive age. Women with physical disabilities are neglected in reproductive and sexual health. Complex health conditions, mobility limitations, negative disability stereotypes, and misconceptions about their needs directly and indirectly affect their sexual and reproductive health. Access to appropriate reproductive health care is crucial for empowering these women to manage their reproductive and sexual health challenges effectively. This scoping review aims to identify, explore, and map the existing evidence on reproductive health programs and interventions in women with physical disabilities.

### Materials and methods

This scoping review will be based on Arksey and O'Malley's methodological framework and the subsequent recommendations of Levac et al. We will search for studies published from January 2006–2024 in English and Persian without restriction by study design in databases such as Web of Science, PubMed, ProQuest, Springer, Google Scholar, ScienceDirect, Cochrane Library, Agency for Healthcare Research and Quality, and websites like the World Health Organization, UNICEF and government sites for policies and reports about reproductive health programs for women with physical disabilities. Two or more reviewers will independently conduct all stages of the study. Data synthesis will utilize a qualitative analysis approach, and this review will be reported using the PRISMA-ScR checklist.

### Conclusion

This scoping review synthesizes evidence on reproductive health interventions for women with physical disabilities, including educational and awareness programs,

**Data availability statement:** No datasets were generated or analysed during the current study. All relevant data from this study will be made available upon study completion.

**Funding:** The author(s) received no specific funding for this work.

**Competing interests:** The authors have declared that no competing interests exist.

healthcare access initiatives, support groups, and policy interventions. We will explore these interventions across various dimensions of reproductive health. The results of this review will offer a comprehensive overview of available resources and identify critical gaps in reproductive health for this population. To our understanding, this study may be the first scoping review in this field, which could provide valuable guidance to key stakeholders.

## Introduction

Today, in many countries of the world, a large share of health concerns is the growing number of people with disabilities [1]. As reported by the World Health Organization (WHO), approximately 15% of the world's people live with disabilities, 15% of whom are women of reproductive age [2]. A person with physical disabilities is defined as someone with limited mobility due to a severe physical impairment, such as paralysis, limb defect, or amputation [3]. In many countries, people with disabilities are among the most deprived and marginalized social groups, who generally have poorer public health, lower educational accomplishments, fewer job opportunities, and higher rates of poverty than people without disabilities [4]. Especially women with disabilities who suffer from double deprivation due to the intersection of gender and disability [5].

In addition, disability is a key determinant of health and influences health policies [3]. International frameworks like The International Conference on Population and Development (ICPD) [6] and the Convention on the Rights of Persons with Disabilities(CRPD) [7] recognize reproductive health (RH) as a basic human right and mandate equitable access to RH information and services for all, regardless of their personal and physical characteristics [8], However, despite these efforts, people with disabilities face many challenges in accessing RH services [9].

Shiwakoti's study revealed that only 15% of women with disabilities utilize RH services, and 81% of respondents attributed their loss of access to these services to inadequate healthcare facilities [10]. Women with disabilities often exhibit low awareness and poor health status before pregnancy [11], and due to inadequate knowledge about contraception methods, they face a higher risk of unintended pregnancies [12]. Research shows that 53% of women with disabilities in the United States [13], and 28.9% in Pakistan [14] experience unintended pregnancies. However, Abadi's findings indicated that only 10.5% of women receiving family planning services had obtained counseling for selecting an appropriate method [5].

Moreover, pregnancy and childbirth pose negative consequences, such as pre-eclampsia, preterm labor, postpartum hemorrhage, etc., for women with disabilities [15,16]. Evidence indicates that fewer than 37% of pregnant women with disabilities receive initial prenatal care, despite its critical role in preventing complications and promoting maternal and fetal outcomes [5]. Also, studies show that people with disabilities receive less sexual education, increasing their vulnerability to sexually transmitted infections (STI) [17,18]. A systematic review showed the HIV prevalence within individuals with disabilities ranged from 1.1% to 29% [19], and STI prevalence

in Ethiopia was reported at 25.3% [20]. These RH problems and challenges in women with disabilities are exacerbated by barriers such as limited access to existing services, low awareness of RH among women, a lack of disability-friendly services, negative attitudes from healthcare providers and communities towards disability and sexual and RH, as well as experiences of violence and abuse [21]. These findings underscore the urgent need for evidence-based programs and targeted interventions across diverse domains of RH. For this purpose, Vaughan developed the Women with Disability Taking Action on Reproductive and sexual health (W-DARE) program. Results demonstrated that after the program's implementation, participants reported increased awareness of sexual and reproductive health and rights, enhanced self-confidence, and greater peer support for accessing services [22]. Also, Hameed's study findings showed that most health promotion interventions for women with disabilities focus only on raising awareness. However, no interventions specifically targeted improving access to maternal health, family planning, and safe abortion services for people with disabilities [23]. Furthermore, the lack of integrated healthcare for women with physical disabilities creates a gap in offering appropriate RH care [24].

Given the rising number of people with physical disabilities [2], developing targeted interventions to improve their RH conditions and address their specific needs is critical [25]. While many studies highlight structural barriers to accessing RH services, adverse health outcomes, and gaps in the service delivery for women with disabilities [10,15,21], most focus only on describing challenges [23]. Few studies explore RH programs and interventions designed specifically for this population. In addition, existing programs are largely confined to awareness campaigns and individual education [23,26] while neglecting essential services tailored to the needs of women with disabilities [23].

Furthermore, the majority of existing evidence comes from upper-middle-income countries and urban settings, with little attention to low-income or rural areas where women face both disability and healthcare disparities. Also, the exclusion or limited involvement of women with disabilities in designing interventions has impeded the development of sustainable solutions [22], and the lack of longitudinal studies makes the long-term effectiveness of programs uncertain [23]. These gaps underscore the need for a scoping review to map existing interventions, identify useful programs, and advance inclusive, rights-based policies. The scoping review is an ideal methodological approach to define the scope and synthesize existing literature on a topic; offering a comprehensive evidence overview [27]. To date, no study has mapped interventions and RH programs specifically designed for women with physical disabilities. This study may be the first scoping review in this field. This study's findings will permit researchers to identify the scope, nature, and gaps in existing research, programs, and interventions related to the RH of women with physical disabilities and establish a foundation for future research. This scoping review aims to identify, explore, and map the available evidence on RH programs and interventions for women with physical disabilities.

## Materials and methods

This study employs a scoping review approach based on Arksey and O'Malley's framework, supplemented by methodological recommendations from Levak et al. and Peters et al. [28–30]. The framework comprises the following steps: (1) identifying the research question; (2) identifying relevant studies; (3) selecting studies; (4) charting data; (5) collating, summarizing, and reporting the results; and (6) consulting stakeholders. The purpose of this scoping review is to map existing programs, clinical guidelines, and studies related to RH interventions for women with physical disabilities in Iran and the world.

### Protocol and registration

The Ethics Committee of Shahroud University of Medical Sciences approved this study (Ref. ID: IR.SHMU. REC.1402.053), and the protocol was registered and published on Open Science Framework (Registration DOI: https://doi.org/10.17605/OSF.IO/EZ43S).

## Identifying the research question

The main question for the research is: What are reproductive health programs and interventions in women with physical disabilities based on scientific sources?

The ancillary questions include:

- What part of RH is focused on?
- What are the challenges of women with physical disabilities in accessing RH services?
- What is the state of RH of women with physical disabilities?
- What are the needs of women with physical disabilities in RH?

**Eligibility of research question.** This study uses the Population-Concept-Context (PCC) framework [31] to determine the suitability of the research question (Table 1).

## Identifying relevant studies

**Search strategy.** A comprehensive search of electronic databases will be conducted by two content experts. The following databases will be included: Web of Science, PubMed, ProQuest, Springer, Google Scholar, ScienceDirect, the Cochrane Library, the Agency for Healthcare Research and Quality, and relevant websites like the World Health Organization (WHO), UNICEF, and government sites for policies and reports about RH programs for women with physical disabilities. Grey literature sources (e.g., Google Scholar, Google, and OpenGrey) will also be explored to enhance information collection. Additionally, the citation lists of all significant material will be manually screened for eligible studies.

The literature search will cover publications in English and Persian from January 2006 to January 2024, with no restrictions on study design. The review will encompass qualitative, quantitative, and mixed-methods studies published in peer-reviewed journals and grey literature, provided they address the research objectives.

The search strategy will utilize controlled vocabulary (e.g., MeSH terms) and free-text keywords across databases to maximize the identification of relevant literature. Key search terms include reproductive health, reproductive and sexual health, physical disability, physical limitation, decreased mobility, immobility, physical-motor handicap, impairment, activity limitation, physical-motor disabilities, program, intervention, care, and women. The initial literature search was performed on 31 January 2024. Non-journal articles have not yet been gathered. The PubMed search strategy is provided in S1 Table. All identified records will be imported into EndNote 9 for duplicate removal and management. The findings will be synthesized into a manuscript following comprehensive data extraction and analysis.

## Study selection

Eligibility criteria were defined to confirm the selected studies contained the necessary data to address the study's objectives.

**Table 1. PCC framework for determination of suitability of review question.**

| Population | Women with physical disabilities |
|---|---|
| Concept | Reproductive health care and programs and interventions in women with physical disabilities |
| Context | All types of studies from all over the world |

**Eligibility criteria.**

Inclusion criteria

- Articles Published in English or Persian from January 2006 to January 2024.

- Studies exclusively focus on women with physical/mobility disabilities.

- Full-text research articles from diverse geographical contexts.

- Studies employing qualitative, quantitative, mixed-methods, and literature review methodologies.

- Studies exploring the experiences of women with physical disabilities regarding RH services.

- Studies focusing on RH programs, interventions, or care frameworks designed for women with physical disabilities.

Exclude criteria

- Studies do not focus on women with physical disabilities/mobility disabilities.

- Studies involving pharmacological/ drug-based interventions.

- Studies not published in English or Persian.

- Duplicate publications, articles with incomplete datasets, experimental protocols, opinions, conference papers, commentaries, editorials, and preliminary-phase studies.

- Studies conducted on non-human subjects.

- Research involving male participants.

- Studies published outside the January 2006 - January 2024 timeframe.

- Articles lacking full-text availability.

The review process will comprise two sequential stages: (1) title and abstract screening and (2) the full-text evaluation. In the first stages, two researchers will independently screen titles and abstracts of articles entered into the EndNote library based on the eligibility criteria. Articles identified as relevant will proceed to full-text retrieval. In the next step, two reviewers will independently assess the full texts of selected articles. Any discrepancies between reviewers at any step will be resolved through discussion or consultation with a third reviewer. In case where unavailable full texts are unavailable, the corresponding authors will be contacted. Reasons for exclusion articles will be systematically documented and reported in the scoping review, following the Preferred Reporting Items for Systematic Reviews and Meta-Analyses Extension for Scoping Reviews (PRISMA-ScR) guidelines [32]. A PRISMA-ScR flow diagram will illustrate the study selection process (S1 Fig).

Reviewers meet at the abstract screening process's start, middle, and end to address ambiguities or discrepancies in study selection. If required, the search approach or eligibility criteria will be refined.

The total number of included and excluded studies will be recorded during the evaluation period. Before screening the articles, a test for accuracy of evaluation will be performed on a randomly selected 20-article subset to ensure team consensus. The assessment process will be fully documented to enable the repetition of the research by others. Following title and abstract screening, full texts of selected studies will be assessed, and studies without eligibility criteria will be excluded.

## Charting data

As Arksey, O'Malley, Peters et al., recommended, that the data extraction process in scoping review involves descriptive analytical techniques to summarize and synthesize evidence [28–30]. According to the Joanna Briggs Institute (JBI)

guidelines and Peter et al. scoping review framework [29], the authors will design a data extraction form to systematically capture study characteristics aligned with the research objectives, such as title, publication year, author name, study type (mixed-methods designs, qualitative, quantitative, e.g., survey, longitudinal studies, randomized controlled trials (RCT),...), study objectives, and key findings (Table 2).

The research team will review the data extraction form before implementation, and necessary changes will be made if needed. The mentioned form will be completed by two authors independently. Any discrepancies will be resolved through consensus-based discussion among reviewers. All records will be stored and managed via Microsoft Excel. Regular team consultations will ensure alignment with study questions and objectives and integrate emergent themes identified during the review process. Article authors will be contacted if necessary to clarify missing, incomplete, or ambiguous data.

The quality of all included studies will be appraised. Two reviewers will independently assess the methodological rigor of each article using the Joanna Briggs Institute (JBI) Quality Appraisal tool. In this scoping review, the quality of studies will be reported descriptively. The appraisal aims not to draw conclusions, or identify effective interventions, or exclude low-quality studies but rather to 1) provide a comprehensive overview of the existing evidence quality and methodological characteristics of the included research, 2) identify critical research gaps in the field, and 3) aid in the interpretation of findings while accounting for potential study limitations.

## Collating, summarizing, and reporting the results

At this stage, guided by Levak et al.'s recommendations [28], the study will focus on three core tasks: data evaluation, results reporting, and interpretation of findings.

A qualitative analysis approach will be utilized for data synthesis, and the results will be presented descriptively. Data categories will be summarized and reported, focusing on RH programs for women with physical disabilities and RH status, RH needs, and RH challenges. The results, including key concepts and categories, may be visualized in tabular or map format.

The results will undergo detailed analysis using a narrative approach to identify gaps in existing research evidence. Comprehensive outcomes will be presented to inform detailed recommendations for future studies. As with any review, the results of this review will be interpreted in light of its methodological limitations. To minimize bias and ensure interpretative consistency, the research team will conduct regular meetings and consensus discussions.

The scoping review protocol and final report will adhere to the PRISMA-ScR guidelines. After completing the data charting, the report's exact format will be finalized as a team.

**Table 2. Variables included in the data extraction form.**

| Author and date |
| --- |
| Study title |
| Objective or research question |
| Sample selection method |
| Research design |
| Data collection (what methods were used to collect data?) |
| Data analysis (what methods were used to analyze the data?) |
| Intervention |
| Result of the Intervention |
| Most relevant findings |
| Author and date |

### Consulting with relevant stakeholders

Arksey and O'Malley suggest that stakeholder consultation enhances the effectiveness of scoping studies [30]. In this study, stakeholders will be purposively sampled from three groups: expertise (researchers in RH and professional academic librarians with ≥5 years of experience), influence (actively practicing reproductive healthcare providers), and lived experience (women with physical disabilities). At least two stakeholders from each group will participate in the study. Stakeholder participation will be individual (in person or remotely) through semi-structured interviews (60–90 minutes).

At the outset of the study, researchers collaborated to define the research question, scope, and eligibility criteria. Additionally, academic librarians provided guidance on search strategies, database selection, search terms, and grey literature sourcing.

Furthermore, to enable a robust interpretation of findings and to identify evidence gaps, preliminary results of the literature review will be disseminated to stakeholders. Their feedback will inform the identification of future research priorities and support the development of evidence-based, actionable recommendations. Stakeholder insights will be thematically analyzed using Braun & Clarke's approach. Identified themes will then be compared and integrated with findings of the literature review to identify alignment, contradictions, and novel insights that may not be evident in the published literature.

Ethical principles will be upheld, including obtaining informed consent from all participants before their involvement. A detailed account of stakeholder engagement will be reported in the review article in alignment with the PRISMA-ScR guidelines. the impact of stakeholder input on outcomes will be illustrated through comparative tables (e.g., pre- and post-consultation revisions).

### Preliminary findings

In the search for studies from the specified databases, 4,778 records were extracted, and the screening process is underway. After completing the screening, data extraction, and synthesis, the findings will be published as a scoping review article.

### Discussion

People with disabilities are among the most socioeconomically deprived groups worldwide [4]. Women with disabilities face harsher conditions than men with disabilities because of their gender [5]. Physical disabilities can significantly impact diverse aspects of an individual's life [15]. Women with physical disabilities are often seen as symbols of incompetence and passivity in society [33], and they may be extra profoundly impacted by disability due to the special conditions associated with menstruation, pregnancy, childbirth, and breastfeeding [34]. Several recent studies indicate that women with disabilities are neglected in many low-income environments, and their reproductive rights are denied [34,35]. Furthermore, evidence shows that a significant number of people with disabilities have restricted knowledge of RH, and many have not received basic RH education [3]. This neglect partly stems from the misconception that individuals with disabilities are not sexually active and are less probably to marry or have children compared to those without disabilities [36]. However, studies show that 91% of people with disabilities in Ghana [37] and 59.9% in Ethiopia [38] have reported having sexual intercourse. Moreover, 15.4% to 67% of women with physical disabilities in Ethiopia experience unintended pregnancies, many of which are terminated through unsafe methods [39]. This underscores the need for education and interventions to improve pregnancy-related healthcare and promote contraceptive use [37]. Additionally, a systematic review indicated that the HIV prevalence among people with disabilities varies from 1.1% to 29% [19]. These findings highlight the urgent need for awareness-raising about reproductive and sexual health for women with disabilities [40]. These challenges are exacerbated by healthcare providers' lack of awareness, limited access to information and services, and communication barriers [41]. This underscores the necessity for specific and integrated programs to provide RH services that meet the needs of women with disabilities [26]. Despite numerous studies highlighting the RH issues faced by women with

disabilities, evidence on programs and interventions specifically designed for the RH of women with physical disabilities remains limited. This gap shows the necessity for a scoping review to identify RH programs and interventions for this population. To our knowledge, this review is the first to focus on identifying RH programs for women with physical disabilities. This scoping review aims to identify and summarize existing evidence on RH programs and interventions for women with physical disabilities. We will also assess the quality of the available studies to provide an overview of the quality of studies and highlight research gaps in this field. The results of this scoping review can benefit RH and disability researchers by helping to identify research gaps and guide future research. Additionally, the findings can provide valuable guidance to key stakeholders related to reproductive healthcare for women with disabilities.

This review follows the methodological framework of Arksey and O'Malley, ensures a transparent process and reliable results, as well as a comprehensive search strategy to identify an extensive variety of studies. Key strengths include the use of two independent reviewers to screen studies. Due to language limitations, only articles fully available in Persian or English will be included.

## Supporting information

**S1 File. PRISMA-P checklist.**
(DOCX)

**S2 File. PRISMA-ScR checklist.**
(DOCX)

**S1 Table. Sample search strategy for PubMed.**
(PDF)

**S1 Fig. Search flowchart following PRISMA-ScR.**
(TIF)

## Author contributions

**Conceptualization:** Maryam Abedi, Mehran Soleymani, Seyedeh Solmaz Talebi, Zahra motaghi.

**Methodology:** Maryam Abedi, Mehran Soleymani, Seyedeh Solmaz Talebi, Zahra motaghi.

**Writing – original draft:** Maryam Abedi, Zahra motaghi.

**Writing – review & editing:** Maryam Abedi, Mehran Soleymani, Seyedeh Solmaz Talebi, Zahra motaghi.

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
