## [Decision Letter · Decision Letter 0]

3 Jan 2025

PONE-D-24-33011Reproductive health programs in women with physical disabilities: a scoping review protocolPLOS ONE

Dear Dr. abedi,

Thank you for submitting your manuscript to PLOS ONE. After careful consideration, we feel that it has merit but does not fully meet PLOS ONE’s publication criteria as it currently stands. Therefore, we invite you to submit a revised version of the manuscript that addresses the points raised during the review process.

The article requires a considerable implementation of structure and methodology as indicated by reviewers.

We look forward to receiving your revised manuscript.

Kind regards,

Andrea Cioffi

Academic Editor

PLOS ONE

Journal Requirements:

3. Please include a caption for figure 1.

Reviewers' comments:

Reviewer's Responses to Questions

**Comments to the Author**

1. Does the manuscript provide a valid rationale for the proposed study, with clearly identified and justified research questions?

Reviewer #1: Partly

Reviewer #2: Yes

Reviewer #3: Yes

Reviewer #4: Partly

2. Is the protocol technically sound and planned in a manner that will lead to a meaningful outcome and allow testing the stated hypotheses?

Reviewer #1: Partly

Reviewer #2: Yes

Reviewer #3: No

Reviewer #4: Partly

3. Is the methodology feasible and described in sufficient detail to allow the work to be replicable?

Reviewer #1: Yes

Reviewer #2: No

Reviewer #3: Yes

Reviewer #4: No

4. Have the authors described where all data underlying the findings will be made available when the study is complete?

Reviewer #1: No

Reviewer #2: Yes

Reviewer #3: No

Reviewer #4: Yes

5. Is the manuscript presented in an intelligible fashion and written in standard English?

Reviewer #1: Yes

Reviewer #2: Yes

Reviewer #3: Yes

Reviewer #4: No

6. Review Comments to the Author

You may also provide optional suggestions and comments to authors that they might find helpful in planning their study.

Reviewer #1: Thank you for the opportunity to review the manuscript "Reproductive health programs in women with physical disabilities: a scoping review protocol". This is a very important topic with potential to improve the reproductive health of a highly marginalized population group in Iran. My enthusiasm for the article is however damped by some major flows including:

1. The very limited search terms the authors plan to use as in line 122. These search terms are very limited. Other terms used for these should be considered e.g impairment, activity limitation, participation limitation, sexual health etc

2. Authors do not intend to do any form of quality assessment. I do not understand why they chose not to do this. A quality assessment is necessary if the authors intend making conclusions, as they have done, on what are appropriate interventions and programs for women with disabilities

3. Authors make very bold conclusions about the implications of their findings such as what are appropriate programs and interventions for women with disabilities. Given the two issues above, such a conclusion cannot be supported. Fundamentally, scoping reviews not ideal for designing programs or interventions. Their primary purpose is to scope the literature to identify gaps in the knowledge, determine feasibility of a systematic review and make recommendations on future research.

Finally, the manuscript can benefit from a through proofreading or making use of the services of a language editor. It is full of grammatical and typographical errors.

Hope these comments will not discourage the authors but rather help them make the manuscript better.

Reviewer #2: Thank you for sending your study protocol on reproductive health programs for women with physical disabilities.

Please find attached my comments for your review:

1- Please rewrite your abstract into subcategories for better clarity.

2- Emphasize the type of study you are conducting.

3- I recommend replacing the term "disabled" with "individuals with special needs".

4- In the introduction, clearly highlight the research gap that your study aims to address.

Reviewer #3: This would be good to resubmit when the project is complete and recommendations/ findings are included.

Reviewer #4: Thank you for giving me the opportunity to read and review this interesting article. It highlights an important area, which is often overlocked.

Abstract:

1. Lack of specificity: The abstract mentions the aim of the scoping review but does not clearly specify the types of reproductive health programs or interventions that will be explored. It could benefit from a more detailed overview of the expected findings.

Introduction:

2. Limited context: The introduction highlights the neglect of reproductive health needs for women with physical disabilities but lacks specific examples of existing programs or studies illustrating the current state of research in this area.

3. Insufficient justification: The rationale for the review’s importance could be strengthened by including statistics or data emphasizing the impact of inadequate reproductive health services on this population.

Methods:

4. Inclusion criteria ambiguity: Mention of including studies with design flaws may lead to inconsistencies in data synthesis. Clearer inclusion/exclusion criteria are required for reliability.

5. Stakeholder consultation details: While the protocol mentions consulting stakeholders, it does not detail how these consultations will be conducted or integrated into the review process.

Results:

6. Lack of preliminary findings: Although it is a protocol, including a section outlining expected outcomes based on preliminary literature could provide context and significance to the review.

7. Reporting framework: The document states results will follow the PRISMA-ScR checklist but does not specify how findings will be categorized or organized.

Discussion:

8. Limited engagement with existing literature: The discussion could better engage with existing studies to contextualize the identified gaps in the literature.

9. Future research directions: While mentioning that the review will guide future research, the document does not specify potential areas for exploration, leaving actionable insights for researchers vague.

10. Clarity and grammatical issues: Various grammar, sentence structure, and language clarity issues need addressing for better readability and professionalism.

7. PLOS authors have the option to publish the peer review history of their article (what does this mean? ). If published, this will include your full peer review and any attached files.

**Do you want your identity to be public for this peer review?** For information about this choice, including consent withdrawal, please see our Privacy Policy .

Reviewer #1: No

Reviewer #2: No

Reviewer #3: **Yes: ** Bassent E Abdelbary

Reviewer #4: No

---

## [Author Response · Author response to Decision Letter 1]

21 Jan 2025

Dear Editor and Reviewers,

Thank you very much for your time and effort in reviewing our manuscript titled "Reproductive health programs in women with physical disabilities: a scoping review protocol" and for providing us with constructive feedback. We greatly appreciate the valuable comments and suggestions, which have helped us improve the quality of our work.

We have carefully addressed all the reviewers' comments and made the necessary revisions to the manuscript. Below, we provide a point-by-point response to each comment and detail the changes made in the revised manuscript.

Reviewer #1

1 ) The very limited search terms the authors plan to use as in line 122. These search terms are very limited. Other terms used for these should be considered e.g impairment, activity limitation, participation limitation, sexual health etc.

answer) Thank you for your valuable feedback regarding the search terms. Based on your comments, we have added the following keywords to our search strategy Reproductive and sexual health, physical limitations, decreased mobility, immobility, Physical-Motor Handicap, impairment, activity limitation, Physical-Motor Disabilities, and program, care. These terms have been included in lines 151-153 of the revised manuscript.

2) Authors do not intend to do any form of quality assessment. I do not understand why they chose not to do this. A quality assessment is necessary if the authors intend making conclusions, as they have done, on what are appropriate interventions and programs for women with disabilities

Thank you for your insightful comment. As the purpose of this scoping review is to map existing programs related to reproductive health for women with disabilities, our primary objective is not to draw conclusions or identify effective programs. However, we fully agree with your suggestion that assessing the included studies' quality can significantly improve our review's transparency and rigor, and provide an overview of the quality of existing studies and help identify research gaps. In response to your constructive input, we have incorporated a quality assessment step after the screening process and will provide an overview of the quality of existing studies. However, as stated, our goal in assessing the quality of studies is not to draw conclusions or select studies based on quality or to identify effective studies.

We sincerely appreciate your thoughtful suggestion, which has undoubtedly strengthened the quality of our work.

The description of the quality assessment process has been added to the Methods section (see lines 224-228).

3) Authors make very bold conclusions about the implications of their findings such as what are appropriate programs and interventions for women with disabilities. Given the two issues above, such a conclusion cannot be supported. Fundamentally, scoping reviews not ideal for designing programs or interventions. Their primary purpose is to scope the literature to identify gaps in the knowledge, determine feasibility of a systematic review and make recommendations on future research.

-Thank you for your careful observation and valuable feedback. We also completely agree with your opinion, and our goal in this scoping review is to identify and summarize studies in the field of reproductive health programs for women with physical disabilities, and we are not looking for conclusions and identification of effective programs. In response to your comment, we have revised the Conclusion section to better align with the goals of a scoping review (see line38-44).

4) Finally, the manuscript can benefit from a through proofreading or making use of the services of a language editor. It is full of grammatical and typographical errors

-Thank you for your valuable feedback. In response to your suggestion, we have carefully reviewed the entire manuscript and utilized the services of a professional language editor to address grammatical and typographical errors.

Please review the revised version, and if further edits are needed, we are happy to make additional adjustments.

Reviewer #2

1) Please rewrite your abstract into subcategories for better clarity.

answer) Thank you for your valuable suggestion to improve the clarity of the abstract. We have rewritten the abstract and divided it into subsections to make its structure clearer and more readable (please see line 20-44).

2) Emphasize the type of study you are conducting.

-Thank you for your valuable feedback. We have revised the manuscript in various sections, such as the introduction and discussion sections, to more clearly emphasize the type of study being conducted) please see line99-102,285-290).

3) I recommend replacing the term "disabled" with "individuals with special needs".

- Thank you for your thoughtful suggestion regarding the terminology used in our manuscript. We appreciate your perspective on replacing the term "disabled" with "individuals with special needs." While we acknowledge the importance of using inclusive and respectful language, we have chosen to retain the term "disabled" for the following reasons:

It is widely recognized and used in scientific, legal, and policy contexts.

It carries specific social and cultural significance, particularly within the disability rights movement, where many individuals and organizations prefer this term as it emphasizes social barriers rather than focusing solely on individual limitations.

If you have specific suggestions for alternative terms or phrases in other parts of the manuscript, we would be happy to consider them.

4) In the introduction, clearly highlight the research gap that your study aims to address

-Thank you for your feedback. We have revised the Introduction by adding sample studies and further explanation to better highlight the research gap (please see lines 61-100).

We appreciate your time and welcome any further suggestions.

Reviewer #3

1) This would be good to resubmit when the project is complete and recommendations/ findings are included.

answer Thank you for your valuable suggestion. We fully agree with your opinion that providing recommendations and findings after the project is completed can help improve the quality and make the study more applicable. Currently, this article is presented as a protocol, but after the project is completed and the data is analyzed, we will submit the final version with full findings and recommendations for re-review.

Thank you for your support and guidance.

Reviewer #4

1) Lack of specificity: The abstract mentions the aim of the scoping review but does not clearly specify the types of reproductive health programs or interventions that will be explored. It could benefit from a more detailed overview of the expected findings.

answer) Thank you for your insightful feedback. In this scoping review, we aim to comprehensively identify reproductive health programs specifically designed for women with disabilities. This will include a wide range of interventions, such as Educational and awareness programs, Initiatives to improve access to healthcare services, support groups and advocacy efforts, and Policy-level interventions. These interventions will be explored across various dimensions of reproductive health.

This content has been added to the abstract section (please see lines 38-41).

2) Limited context: The introduction highlights the neglect of reproductive health needs for women with physical disabilities but lacks specific examples of existing programs or studies illustrating the current state of research in this area

-Thank you for your valuable feedback. We fully agree with your comment that providing specific examples of existing programs or studies can help to better understand the current state of research in this area. In response to this suggestion, we have revised the Introduction section and added examples of programs and studies relevant to the reproductive health of women with physical disabilities (please see lines 61-91).

3) Insufficient justification: The rationale for the review’s importance could be strengthened by including statistics or data emphasizing the impact of inadequate reproductive health services on this population

-Thank you for your feedback. We have strengthened the review’s rationale by adding statistics and data related to adverse services and reproductive health problems of women with physical disabilities. These updates are included in the Introduction (please see lines 61-82).

4) Inclusion criteria ambiguity: Mention of including studies with design flaws may lead to inconsistencies in data synthesis. Clearer inclusion/exclusion criteria are required for reliability

-Thank you for your feedback. we have revised the inclusion and exclusion criteria to provide clearer and more detailed guidelines. These updates can be found in the Methods section (please see line 166-184).

5) Stakeholder consultation details: While the protocol mentions consulting stakeholders, it does not detail how these consultations will be conducted or integrated into the review process.

-We appreciate your suggestion to provide more details on how these consultations will be conducted and integrated into the review process. In response, we have added a detailed description of the stakeholder consultation process (please see line 248-255).

6) Lack of preliminary findings: Although it is a protocol, including a section outlining expected outcomes based on preliminary literature could provide context and significance to the review.

-Thank you for your valuable suggestion regarding the inclusion of preliminary findings. As we are currently in the screening phase, we have incorporated the preliminary findings from the study search into the manuscript. (Please see line 256-259).

7) Reporting framework: The document states results will follow the PRISMA-ScR checklist but does not specify how findings will be categorized or organized

Thank you for your feedback. We have clarified the categorization of findings according to the PRISMA-ScR checklist. Results will be organized into:

1. Reproductive health programs,

2. Reproductive health status,

3. Reproductive health needs, and

4. Reproductive health challenges for women with physical disabilities.

These categories will ensure a structured and comprehensive presentation of the findings, aligning with the objectives of the review. We have updated the Methods section to reflect this organization (please see lines 234-237).

8) Limited engagement with existing literature: The discussion could better engage with existing studies to contextualize the identified gaps in the literature

-We sincerely appreciate your valuable suggestion. We fully agree that greater engagement with existing studies can enhance the understanding of gaps in the literature. In response to your feedback, we have revised the Discussion section by incorporating additional references (please see the Discussion section, line 266-281).

9) Future research directions: While mentioning that the review will guide future research, the document does not specify potential areas for exploration, leaving actionable insights for researchers vague.

-We are deeply grateful for your insightful feedback. we have enriched the Discussion section by incorporating additional content aimed at resolving the identified issues (please see the Discussion section, line 266-295 ).

10) Clarity and grammatical issues: Various grammar, sentence structure, and language clarity issues need addressing for better readability and professionalism

-Thank you for your feedback on grammar and clarity issues. We have revised the manuscript to improve sentence structure, grammar, and readability and utilized the services of a professional language editor to address grammatical and typographical errors.

We appreciate your time and welcome any further suggestions.

We have incorporated all the suggested revisions and hope that the revised manuscript now meets the journal's standards. Should you require any further clarification or additional information, please do not hesitate to contact us.

Once again, we sincerely thank you and the reviewers for your time and valuable input. We look forward to hearing from you regarding the next steps.

Best regards,

Maryam Abedi

---

## [Decision Letter · Decision Letter 1]

24 Apr 2025

PONE-D-24-33011R1Reproductive health programs in women with physical disabilities: a scoping review protocolPLOS ONE

Dear Dr. Motaghi,

Thank you for submitting your manuscript to PLOS ONE. After careful consideration, we feel that it has merit but does not fully meet PLOS ONE’s publication criteria as it currently stands. Therefore, we invite you to submit a revised version of the manuscript that addresses the points raised during the review process.

We look forward to receiving your revised manuscript.

Kind regards,

Andrea Cioffi

Academic Editor

PLOS ONE

Reviewers' comments:

Reviewer's Responses to Questions

**Comments to the Author**

1. Does the manuscript provide a valid rationale for the proposed study, with clearly identified and justified research questions?

Reviewer #2: Yes

2. Is the protocol technically sound and planned in a manner that will lead to a meaningful outcome and allow testing the stated hypotheses?

Reviewer #2: Yes

3. Is the methodology feasible and described in sufficient detail to allow the work to be replicable?

Reviewer #2: Yes

4. Have the authors described where all data underlying the findings will be made available when the study is complete?

Reviewer #2: No

5. Is the manuscript presented in an intelligible fashion and written in standard English?

Reviewer #2: No

6. Review Comments to the Author

You may also provide optional suggestions and comments to authors that they might find helpful in planning their study.

Reviewer #2: Thank you for the previous revision and your hard work to address the previous reviewers' comments.

1- The research gap still not clear.

2- The authors have chosen to retain the term "disabled" rather than using "individuals with special needs." While this is justified, the manuscript should consistently use person-first language (e.g., "women with disabilities" rather than "disabled women")

3- The authors mention using the Joanna Briggs Institute (JBI) Quality Appraisal tool but do not specify how the results of this assessment will be used. Will low-quality studies be excluded, or will the quality assessment simply inform the interpretation of findings? This needs to be clarified.

4- Some sections of the manuscript, particularly the methods, are still difficult to follow due to overly complex sentence structures. Also some minor language defects, I suggest proof reading by a native English.

5- how stakeholders will be selected and how their input will be integrated.

7. PLOS authors have the option to publish the peer review history of their article (what does this mean? ). If published, this will include your full peer review and any attached files.

**Do you want your identity to be public for this peer review?** For information about this choice, including consent withdrawal, please see our Privacy Policy .

Reviewer #2: No

---

## [Author Response · Author response to Decision Letter 2]

7 May 2025

In response to your comments:

We have carefully considered each point and incorporated revisions to address them comprehensively. Below, we provide a detailed, point-by-point response to your feedback, along with the specific changes made in the revised manuscript (highlighted in track changes or noted by line/page numbers).

1) The research gap still not clear.

Thank you for your careful attention to this important point. In the revised introduction section (lines 89-107), we have clarified the research gap by adding examples of key limitations, including:

• Lack of evidence on interventions to improve access to reproductive health services for women with physical disabilities in underprivileged areas,

• Dominant focus of existing studies on challenges and barriers faced by women, without addressing health promotion programs,

• Scarcity of longitudinal studies to assess the sustainability and long-term impact of interventions,

• Inequity in existing data, with a predominance of evidence from high-income regions and neglect of rural/remote communities.

We hope these revisions more accurately reflect unaddressed research gaps in the reproductive health of women with physical disabilities. We sincerely appreciate your insightful feedback and welcome any additional suggestions to further improve this work.

2) The authors have chosen to retain the term "disabled" rather than using "individuals with special needs." While this is justified, the manuscript should consistently use person-first language (e.g., "women with disabilities" rather than "disabled women")

We sincerely appreciate the reviewer’s insightful feedback regarding the use of person-first language. We fully align with the importance of emphasizing individuality and dignity by using terms like “women with disabilities” instead of “disabled women”. In response to this guidance, we have meticulously revised the entire manuscript to ensure consistent application of person-first language.

Thank you for this valuable suggestion, which has strengthened the manuscript’s inclusivity and precision.

3) The authors mention using the Joanna Briggs Institute (JBI) Quality Appraisal tool but do not specify how the results of this assessment will be used. Will low-quality studies be excluded, or will the quality assessment simply inform the interpretation of findings? This needs to be clarified.

Thank you for your insightful comments. As the purpose of this scoping review is to map existing reproductive health programs for women with disabilities, our primary goal is not to draw conclusions or identify effective programs or select studies based on quality or exclude low-quality studies. Our goal is simply to:

• Provide a comprehensive overview of the existing evidence quality and methodological characteristics of the included research,

• Identify critical research gaps in the field, and

• Aid in the interpretation of findings while accounting for potential study limitations.

The results of this assessment will be fully included in the final article.

Thank you again for this important point, which greatly contributed to the article's clarity.

Please see the changes related to this section on lines 221-227.

4) Some sections of the manuscript, particularly the methods, are still difficult to follow due to overly complex sentence structures. Also, some minor language defects, I suggest proof reading by a native English.

Thank you for your meticulous feedback on improving the manuscript’s clarity and language quality. We deeply value your expertise and have implemented the following revisions to address your concerns:

• Complex sentences in the Methods section (e.g., search strategy, inclusion criteria) have been restructured into shorter, step-by-step descriptions

• The entire manuscript underwent rigorous grammatical and stylistic revisions to resolve ambiguities and improve flow.

• A native English-speaking editor (certified by NativeEnglishEdit.com, with REFER CODE:EE-2024-333326096- ABEDI-2870 NEW 1122 REV.2) proofread the final version to ensure linguistic precision and natural phrasing.

We believe these revisions align the manuscript with the journal’s standards for clarity and rigor. Should any sections still require refinement, we are fully committed to further adjustments.

Thank you again for your invaluable guidance in strengthening this work.

5) how stakeholders will be selected and how their input will be integrated.

We sincerely appreciate the reviewer’s insightful comment regarding stakeholder engagement. In response, we have elaborated on both the stakeholder selection process and the strategies for integrating their input in the revised manuscript. These details are now included in the section titled "Consulting with Relevant Stakeholders" (Lines 244–264).

Thank you again for your thoughtful feedback, which helped us improve the clarity and completeness of this section.

---

## [Decision Letter · Decision Letter 2]

8 Jul 2025

Reproductive health programs in women with physical disabilities: a scoping review protocol

PONE-D-24-33011R2

Dear Dr. motaghi,

We’re pleased to inform you that your manuscript has been judged scientifically suitable for publication and will be formally accepted for publication once it meets all outstanding technical requirements.

Kind regards,

Udoka Okpalauwaekwe, MD, MPH, PhD

Academic Editor

PLOS ONE

Additional Editor Comments (optional):

Reviewers' comments:

Reviewer's Responses to Questions

**Comments to the Author**

1. Does the manuscript provide a valid rationale for the proposed study, with clearly identified and justified research questions?

Reviewer #1: Yes

Reviewer #2: Yes

2. Is the protocol technically sound and planned in a manner that will lead to a meaningful outcome and allow testing the stated hypotheses?

Reviewer #1: Yes

Reviewer #2: Yes

3. Is the methodology feasible and described in sufficient detail to allow the work to be replicable?

Reviewer #1: Yes

Reviewer #2: Yes

4. Have the authors described where all data underlying the findings will be made available when the study is complete?

Reviewer #1: No

Reviewer #2: Yes

5. Is the manuscript presented in an intelligible fashion and written in standard English?

Reviewer #1: Yes

Reviewer #2: Yes

6. Review Comments to the Author

You may also provide optional suggestions and comments to authors that they might find helpful in planning their study.

Reviewer #1: Thank you for the opportunity to reviewer this manuscript again. Authors have adequately addressed all reviewer comments. The manuscript flow and readability is much improved after the English language editorial.

Reviewer #2: Thank you for addressing previous reviewer's comments. At this stage, I would accept that this protocol is suitable for publication. Best regards

7. PLOS authors have the option to publish the peer review history of their article (what does this mean? ). If published, this will include your full peer review and any attached files.

**Do you want your identity to be public for this peer review?** For information about this choice, including consent withdrawal, please see our Privacy Policy .

Reviewer #1: **Yes: ** Makandwe Nyirenda

Reviewer #2: No

---

## [Editor Report · Acceptance letter]

PONE-D-24-33011R2

PLOS ONE

Dear Dr. motaghi,

I'm pleased to inform you that your manuscript has been deemed suitable for publication in PLOS ONE. Congratulations! Your manuscript is now being handed over to our production team.

Kind regards,

on behalf of

Dr. Udoka Okpalauwaekwe

Academic Editor

PLOS ONE